# Dissipation Behavior and Acute Dietary Risk Assessment of Thiamethoxam and Its Metabolite Clothianidin on Spinach

**DOI:** 10.3390/molecules27072209

**Published:** 2022-03-29

**Authors:** Yanmei Yang, Shu Qin, Xia Wang, Junli Cao, Jindong Li

**Affiliations:** Shanxi Center for Testing of Functional Agro-Products, Shanxi Agricultural University, Taiyuan 030031, China; ymyangsuda@163.com (Y.Y.); qinshu55@126.com (S.Q.); wangxialiao@126.com (X.W.); caojunli@sxau.edu.cn (J.C.)

**Keywords:** thiamethoxam, clothianidin, spinach, LC-MS/MS, dissipation behavior, acute dietary risk assessment

## Abstract

Thiamethoxam and its metabolite clothianidin residues pose a potential threat to human health. This study aims to investigate the residue behavior and acute dietary risk assessment of thiamethoxam and clothianidin on spinach. Thiamethoxam and clothianidin were extracted using a quick, easy, cheap, effective, rugged, safe (QuEChERS) method and analyzed using liquid chromatography-tandem mass spectrometry (LC-MS/MS). At spike levels from 0.01 to 5 mg kg^−1^, the average recoveries of both analytes were in the range of 94.5–105.5%, with relative standard deviations (RSDs) of 3.8–10.9%. The dissipation behavior of thiamethoxam followed first-order kinetics, with half-lives of ≤1.6 days. Clothianidin appeared readily as a plant metabolite with highest level exhibited during 3 to 5 days after application. Temperature and light may be two main factors for degradation of thiamethoxam. Besides, acute risk assessment of thiamethoxam and clothianidin was evaluated with risk quotients (RQs) <100%, which suggested a low health risk for all consumer groups of Chinese residents.

## 1. Introduction

As an important leafy vegetable, spinach is rich in calcium, iron, vitamin C, carotenoids, and folic acid with consumer demand increasing annually [1,2]. To ensure the high yields of spinach, the application of pesticides for the control of pests is an indispensable measure in the planting process. Thiamethoxam, [(EZ)-3-(2-chloro-1,3-thiazol-5-ylmethyl)-5-methyl-1,3,5-oxadiazinan-4-ylidene(nitro)amine] is a neonicotinoid insecticide, having been widely applied on vegetables to wipe out sucking insect pests by inhibiting nicotinic acetylcholine receptors [3,4]. Approximately 378 commercial formulations of thiamethoxam are registered in China. Clothianidin, (E)-1-(2-chloro-1,3-thiazol-5-ylmethyl)-3-methyl-2-nitroguanidine) is also a neonicotinoid insecticide and a major metabolite of thiamethoxam with similar structure and insecticidal activity [5,6]. In spite of their merits to increase production, the existence of thiamethoxam and clothianidin could cause neurobehavioral alterations in mammals, posing a risk to consumers’ health [7,8]. Furthermore, thiamethoxam is applied directly on edible portions of spinach while the pre-harvest interval of spinach is short. Thus, it is essential to evaluate acute dietary risk of thiamethoxam and clothianidin on spinach. According to the Joint FAO/WHO Meeting on Pesticide Residues (JMPR) report, the residue definition is to consider thiamethoxam and clothianidin separately [9].

To date, many studies have been carried out on the dissipation behavior and dietary risk assessment of thiamethoxam and clothianidin on different matrices, including gogi berry [10], cowpea [11,12], strawberry [13], apple [14,15], citrus [16], mango [17], wheat [18], lettuce [18], and tomato [18]. For example, Rahman et al. found that half-lives on Swiss chard were in reverse order of initial deposition [3]. Li et al. reported that initial concentrations of thiamethoxam in goji berry ranged from 0.297 to 0.720 mg kg^−1^ with half-lives from 1.01 to 1.08 days, demonstrating rapid degradation of thiamethoxam [10]. Fan et al. investigated the dissipation behavior and dietary risk assessment of thiamethoxam on apple. The half-lives of thiamethoxam ranged from 7.01 to 11.9 days, and negligible risk for general consumers was exhibited [14]. In these studies, the difference of original amount and degradation of thiamethoxam was attributed to rainfall, sunlight, organic matter content, temperature, and plant variety et al. However, only the dissipation pattern of parent thiamethoxam was investigated in most studies and factors for the different residue behavior of thiamethoxam have not been studied systematically. Furthermore, there are limited published studies concerning the residue behavior of thiamethoxam on spinach. For example, Zhao et al. investigated that decline dynamic of thiamethoxam in spinach was in accordance with first-order kinetics, and the decline rate of thiamethoxam in green houses was slower than open field. Liu et al. reported that half-life of thiamethoxam on spinach was 2.3 d. Nevertheless, both studies didn’t concern the residue behavior of clothianidin and dietary risk assessment [19,20].

This study aimed to (1) determine residue levels of thiamethoxam and clothianidin on spinach by the QuEChERS method, coupled with LC-MS/MS, (2) investigate residue behavior of thiamethoxam and clothianidin, as well as the main factors for their dissipation under different planting conditions, (3) assess the acute dietary risk of thiamethoxam and clothianidin for different consumer groups in China. The obtained results would help to provide guidance for proper use of thiamethoxam on spinach and dietary safety of spinach consumption.

## 2. Results and Discussion

### 2.1. Extraction and Purification

According to the Guideline on Pesticide Residue Trials (NY/T 788–2018) published by Ministry of Agriculture, China [21], the matrix effects (ME), linearity, sensitivity, accuracy, and precision were investigated to validate the analytical method in this study.

ME of thiamethoxam and clothianidin in spinach were evaluated by comparing the slope of matrix matched standard curve with the slope of solvent standard curve. As shown by Appendix A, ME (%) value of thiamethoxam and clothianidin were −27% and −52%, respectively, indicating significant matrix effects [22]. In order to eliminate matrix influence and ensure accurate quantification, matrix-matched calibration curves were prepared. Representative chromatograms (multiple reaction monitoring, MRM mode) of thiamethoxam and clothianidin on spinach are displayed in Appendix A. The results of regression analyses for thiamethoxam and clothianidin on spinach extracts are shown in Appendix A. In the range from 5 to 200 μg kg^−1^, the correlation coefficients (R^2^) of thiamethoxam and clothianidin were 0.9975 and 1.0000, indicating good linearity when using matrix-matched calibration curves.

The accuracy and precision of the method was evaluated by recoveries and relative standard deviations (RSDs). Mixed standard solutions of thiamethoxam and clothianidin were spiked into blank spinach samples at four concentrations of 0.01, 0.1, 2, and 5.0 mg kg^−1^ with each level repeated five times. Table 1 shows the average recoveries and RSDs of intra-day (*n* = 5) and inter-day (*n* = 15) recovery experiments conducted in three different days. The intra- and inter-day accuracy was in the range 94.5–98.8% and 96.8–105.1% for thiamethoxam, and 96.3–105.5% and 100.6–109.7% for clothianidin, respectively. Intra- and inter-day precision of the method were ≤10.9% and ≤9.7% for thiamethoxam, and ≤9.3% and ≤8.7% for clothianidin.

The limit of quantitation (LOQ) of both analytes, defined as the lowest spiked concentration with S/N higher than 10, were 0.01 mg kg^−1^.

Based on the above results, it is feasible to determine thiamethoxam and clothianidin on spinach using the developed QuEChERS method, which was applied for the detection of field samples in the following experiment.

### 2.2. Dissipation Study

Dissipation trials were performed on spinach samples from Shanxi, Shandong, Anhui, and Guangdong. Before application, the formulation was checked in a laboratory via serials dilution 20,000 times with acetonitrile. The concentration of thiametoxam was 0.119%, which was consistent with data provided by suppliers. The results of dissipation trials are exhibited in Figure 1 and Appendix A. At the Shanxi site, the original amount of thiamethoxam deposited on spinach was 1.3 mg kg^−1^, which was relatively high among four sites, while the original amount in the Anhui and Guangdong sites were 0.81 and 0.79 mg kg^−1^, respectively. At the Shandong site, the original amount of thiamethoxam deposited on spinach was 0.36 mg kg^−1^, which was about two times lower than the Shanxi site. It was noted that original amount of pesticide deposited on plants were affected by various factors, including crop variety, plant height, or density, application machine, weather condition, cultivation facilities, et al. [23]. Although field trials at all sites were carried out under good agricultural practice (GAP), the original amount of thiamethoxam deposited on spinach was different. Dissipation behavior of thiamethoxam in four sites was investigated. For Shanxi, Anhui, and Guangdong sites, the dissipation of thiamethoxam on spinach samples followed a first-order kinetic model (R^2^ = 0.914–0.989) with first rapid and then slow dissipation trends (Table 2 and Appendix A). The half-lives were in the range from 1.3 to 1.6 days. The results were similar to the half-lives of thiamethoxam in tomato (2.34 days) [24], goji berry [10] (1.01–1.08 days), cowpea [11] (0.8–1.6 days), and citrus [16] (1.9 days), reported by previous studies. For the Shandong site, the dissipation rate of thiamethoxam was more rapid than at the other three sites. The amount of thiamethoxam on spinach decreased by 97% after 3 days of application and reached a level lower than LOQ after 5 days of application.

Thiamethoxam degraded fast in crops, and clothianidin appeared readily as a plant metabolite [9,11], as suggested by Figure 1. The amount of metabolite clothianidin on spinach after 2 h of application ranged from 0.092 to 0.51 mg kg^−1^. At Shanxi, Anhui, and Guangdong sites, clothianidin residue increased first and then decreased. During 3 to 5 days after application of thiamethoxam, the highest amount of clothianidin residue was observed. At the Shandong site, the clothianidin residue on spinach after 2 h of application was even higher than for parent thiamethoxam (Figure 1d). According to a report by Li et al., the half-lives of thiamethoxam under ultraviolet B (UVB) and sunlight were 3 and 10 h, respectively, indicating fast degradation rate of thiamethoxam without soil [25]. The loss of thiametoxam was reported to be directly related to the formation of clothianidin [26]. Li et al. also investigated that thiamethoxam degraded much faster without soil than with soil, which was attributed to the block of light penetrating into the samples by soil. In our tests, thiametoxam was applied on spinach by foliar spray, and thiamethoxam were directly exposed to the sunlight without soil, indicating fast degradation of thiamethoxam. Besides, the application date at the Shandong site was July, and the spinach was cultivated in open fields, which indicated the strongest light irradiation among four test sites. Furthermore, the temperature at the time of application at the Shandong site was 30 °C, which was almost 11 °C higher than Anhui sites with the same cultivation facilities. The above factors resulted in fast degradation of thiamethoxam and formation of clothianidin at the Shandong sites.

Crop species, pH value, rain, light intensity, and temperature were reported to play significant roles in affecting the degradation of thiamethoxam in crops [11]. In the following discussion, factors for degradation of thiamethoxamon spinach were investigated. Table 3 summarizes residue of thiamethoxam and corresponding experiment conditions of four test sites during 3 days after application. On the one hand, the higher temperature resulted in a faster degradation rate, as suggested by the comparison between Shandong and Anhui, as well as Shanxi and Guangdong with same cultivation facilities. On the other hand, the degradation rate was also affected by three cultivation facilities, including in a greenhouse, an open field, or in a shaded area. In spite of similar temperature, the degradation rate from Shanxi was lower than that from Anhui, and degradation rate from Guangdong was also lower than Shandong. The lower degradation rates were attributed to the fact that cultivation facilities can reduce light on crops effectively. In conclusion, temperature and light were two main factors for residue behavior of thiamethoxam, which was consistent with other studies [27,28]. The higher temperature and additional light led to faster degradation of thiamethoxam.

### 2.3. Terminal Residues

To reflect the highest residue levels of thiamethoxam and its metabolite clothianidin in spinach, a terminal study was conducted after application once with a recommend highest dosage of 27 g active ingredients per ha (g a.i. ha^−1^) [29,30]. The terminal residues of thiamethoxam and clothianidin in 8 field sites are analyzed and summarized in Table 4. The mean residue levels of thiamethoxam and clothianidin were in the range of <0.010–0.39 mg kg^−1^ and 0.021–0.48 mg kg^−1^, respectively, which were lower than the maximum residue levels (MRLs) prescribed by China [31]. It is noted that the clothianidin residue was higher than parent thiamethoxam, except for the Hunan site. The higher clothianidin residue (in combination with its higher toxicity) than thiamethoxam may present higher public health concern. Thus, it is essential for estimation of the dietary exposure risk.

### 2.4. Acute Dietary Exposure Risk Assessment

According to the report of JMPR, the dietary exposure risk of thiamethoxam and clothianidin were assessed separately [9].

Acute dietary risk assessment for spinach consumption was performed by comparing the international estimated short-term intake (IESTI) with the acute reference dose (ARfD). Based on the terminal residue experiments, the highest residue values (HR) of thiamethoxam and clothianidin on spinach were 0.41 and 0.52 mgkg^−1^, respectively. The edible portion of the unit weight (U) of spinach (WHO) is 90g [32]. The ARfD of thiamethoxam and clothianidin from JMPR report are 1.0 and 0.6 mg (kg bw^−1^), respectively [9,33]. The body weight and spinach intake of consumers with different ages in China are summarized in Table 5 [34,35,36,37].

According to the Formulas (4) and (5) in Section 3.6.3, the IESTI and acute dietary exposure risk probability (RQ_a_%) of thiamethoxam and clothianidin for different consumer groups were calculated, which are exhibited in Table 5 and Figure 2. In general, children were considered to be the most vulnerable group because they were particularly susceptible to the hazards related with pesticides [38]. As shown, the acute dietary intake risks of consumers at different ages varied, and the RQ_a_% values decreased with the increase of age from 3 to 44 years, indicating that thiamethoxam and clothianidin posed more acute dietary risks to children than adults. However, RQ_a_% values of thiamethoxam and clothianidin for all consumer groups were in the range of 0.21–0.73% and 0.44–1.54%. All the RQ_a_% values were <100%, representing that the acute dietary risk after application of thiamethoxam in spinach at a recommended dose and shortest pre-harvest interval (PHI) were low and thus acceptable to Chinese consumers [39].

## 3. Material and Methods

### 3.1. Reagents and Chemicals

The standards of thiamethoxam (99.6%) and clothianidin (99.9%) were supplied by Cato Research Chemicals Inc (Eugene, OR, USA). The 0.12% thiamethoxam, formulated as emulsion, oil in water (EW), was provided by Shenyang Jinzhao Biotechnology Co., Ltd. HPLC-grade acetonitrile (ACN) was purchased from Tedia Company (Fairfield, OH, USA). LC/MS-grade formic acid and methanol were obtained from Fisher Regent Company (Beijing, China) and Merck (Darmstadt, Germany), respectively. Primary secondary amine (PSA, 40–60μm) and graphitized carbon black (GCB, 120–400 Mesh) were from Bonna-Agela Technologies Ltd. (Tianjin, China). Sodium chloride (NaCl) and anhydrous magnesium sulfate (MgSO_4_) were from Sinopharm Chemical Reagent Co., Ltd. (Beijing, China). Nylon syringe filters (0.22 µm) were provided by Shimadzu (China).

Stock standard solution of thiamethoxam (1000 mg L^−1^) and clothianidin (1000 mg L^−1^) were both prepared by dissolving 10 mg of standards with acetonitrile to 10 mL. The mixed standard solution (100, 10, 1 mg L^−1^) was prepared by serially diluting the appropriate amounts of the standard stock solutions with HPLC-grade acetonitrile. Working standard solutions of 0.2, 0.1, 0.05, 0.02, 0.01, and 0.005 mg L^−1^ were prepared by serial dilution of the mixed standard solution with control extracts of spinach. All standard solutions were stored in a freezer at −18 °C before use.

### 3.2. Field Trial and Sample Collection

According to the Guideline on Pesticide Residue Trials (NY/T 788–2018) and Pesticide Registration Residue Test Area Guide issued by Ministry of Agriculture, P. R. China, field trials were implemented, which included a dissipation study and a terminal residue study. Dissipation study was to investigate the dissipation behavior of thiamethoxam. The terminal study was conducted to study residual concentrations of thiamethoxam, based on which acute dietary intake is calculated [16]. 8 test sites were located in Hohhot city in Inner Mongolia Autonomous Region, Taiyuan city in Shanxi province (dissipation), Beijing city, Jinan city in Shandong province (dissipation), Suzhou city in Anhui province (dissipation), Changsha city in Hunan province, Guiyang city in Guizhou province, and Foshan city in Guangdong province (dissipation). The cultivation facilities and climate types of the field sites are exhibited in Appendix A. The treatment was carried out in two plots with an area of not less than 50 m^2^ per plot (one for thiamethoxam application and one for control). The two plots were separated by a buffer area to avoid cross-contamination.

0.12% thiamethoxam, formulated as an EW, was sprayed once with a recommend high dosage of 27 g active ingredient per ha (g a.i.ha^−1^). Before application, the sprayer was tested and thoroughly cleaned. Detailed information about the application process was provided by Appendix A. There was no rain at 2 h after application. The temperature ranged from 18–30 °C and the wind speed was ≤2.0 m s^−1^. Two independently composited samples were collected at each plot. For each sample, a minimum of 12 plants (2.0 to 2.3 kg) with disease-free and normal growth were randomly collected. The dissipation study was based on samples collected at 2 h, 3, 5, 7, and 10 days after application, and the terminal residue study was based on samples collected at 5 and 7 days after application.

The field collected samples were cut into pieces and mixed thoroughly. Two identical laboratory samples (minimum of 200 g) were retained after quartering, one for testing and the other as backup. The samples were transferred in labeled sample bag and stored at −18 °C until analysis.

### 3.3. Sample Preparation

10.0 g of homogenized spinach was weighed into 50 mL PTFE centrifuge tubes. Then, 10 mL of acetonitrile was added and vortexed for 10 min at 2500 rpm. Sequentially, 5.0 g NaCl was added and vortexed for another 5 min. The supernatant was collected by centrifugation at 8000 rpm for 5 min.

For the clean-up step, an aliquot (1.5 mL) of the supernatant was transferred into a 2.0 mL centrifuge tube with 142.5 mg MgSO_4_, 20 mg PSA, and 30 mg GCB. The mixture was vortexed at 2500 rpm for 5 min and centrifuged at 5000 rpm for 2 min. After filtration through a 0.22 µm syringe filter, an aliquot of the supernatant (0.5 mL) was mixed thoroughly with 0.5 mL water for LC–MS/MS analysis.

### 3.4. LC-MS/MS Analysis

The analysis of thiamethoxamand clothianidin was conducted on UPLC I-Class XEVO TQ-S (Waters, Milford, MA, USA), which was coupled to an electrospray ionization source (ESI). The separation of two components was achieved by ACQUITY BEH C18 column (2.1 × 50 mm, 1.7 μm) and mobile phase consisting of 0.1% formic acid in water (phase A) and methanol (phase B). Detailed information is exhibited in Appendix A.

### 3.5. Method Validation

Based on SANTE guidelines, the analytical method was validated concerning matrix effects (ME), linearity, accuracy, precision, and limit of quantitation (LOQ) [40]. In detail, control spinach samples were spiked with thiamethoxam and clothianidin at four fortification levels of 0.01, 0.1, 2 and 5 mg/kg. Five replicates were performed at each spiked level. Then, these spiked samples were placed at ambient temperature for approximately 30 min and processed using the method described in Section 3.3. It is noted that the mixture from spiked samples at 2 and 5 mg kg^−1^ were further diluted 20 and 50 times by acetonitrile, respectively.

Intra-day precision was evaluated by conducting five replicates of each spiked concentration, and inter-day precision was characterized by analyzing five replicates of each spiked concentration daily for three days.

### 3.6. Theoretical Calculations

#### 3.6.1. Matrix Effect (ME)

The ME (%) was calculated by the Equation (1):(1)$$\mathrm{ME}\left(\%\right)\;=100\;\times \;(\frac{\mathrm{Slope}\;\mathrm{of}\;\mathrm{matrix}\;\mathrm{matched}\;\mathrm{standard}\;\mathrm{curve}}{\mathrm{Slope}\;\mathrm{of}\;\mathrm{solvent}\;\mathrm{standard}\;\mathrm{curve}}-1)$$

According to ME (%) value, the matrix effect was divided into two cases. If ME value ranged from −20% to 20%, it is considered that matrix effect can be ignored. If ME (%) value was beyond the listed regions, it was considered to be a significant matrix effect [22].

#### 3.6.2. Dissipation Kinetic

The dissipation curves of thiamethoxam and its metabolite on spinach were simulated with first-order kinetic Equation (2):C_t_ = C_0_e^−kt^(2)
where C_0_ and C_t_ denote concentration of pesticide residue (mg kg^−1^) at 0 and t days, respectively, *k* is the dissipation rate constant.

Half-life (t_1/2_) of thiamethoxam and clothianidin on spinach was determined by Equation (3):t_1/2_ = ln2/k(3)

#### 3.6.3. Acute Dietary Risk Assessment

According to Guidance for International Estimated Short-term Intake (IESTI), the acute dietary risk assessment was conducted using case 1, 2a, 2b and 3, depending on food commodity [41]. Unit weight of spinach is above 25 g, indicating that the meal-sized portion (such as a single piece of spinach) might have a higher residue than the composite. Furthermore, the unit weight of spinach (90 g) is also less than large-portion weight of different consumer groups in this study. The assessment of acute dietary risk assessment was conducted using case 2a. IESTI and acute dietary exposure risk probability (RQ_a_) were evaluated by the following equations:(4)$$\mathrm{IESTI}=\frac{U\;\times \;\mathrm{HR}\;\times \;\nu +(\mathrm{LP}-U)\;\times \;\mathrm{HR}}{\mathrm{bw}}$$
(5)$${\mathrm{RQ}}_{a}\%=\frac{\;\mathrm{IESTI}}{\mathrm{ARfD}\;}\;\times \;100$$

U is the edible portion of the unit weight, kg; HR is the highest residue from supervised trials data, in mg kg^−1^; LP represents highest large portion provided (97.5th percentile of eaters), kg day^−1^; bw is body weight, kg; ν represents variability factor, which was found to be 3 according to the report of International Programme on Chemical Safety in 2009 [42]; Arfd is the acute reference dose from JMPR report [9,33].

## 4. Conclusions

In this study, the residue of thiamethoxam and its metabolite clothianidin on spinach after field application were determined by the QuEChERS method, coupled with LC-MS/MS. The degradation of thiamethoxam follows first-order kinetics on spinach, with t_1/2_ values ranging from 1.3 to 1.6 days. By comparing dispassion behavior and experiment conditions of four test sites, temperature and light might be two main factors for degradation of thiamethoxam. Clothianidin appeared readily as a metabolite, and the highest amount was exhibited during 3–5 days after application of thiamethoxam. Furthermore, the result of terminal residue study indicated that clothianidin residue was higher than parent thiamethoxam at most test sites. The acute dietary risks to different consumer groups in China were also studied. The values of RQ_a_% values were far less than 100%, indicating that no significant risk on human health was posed at the recommended dose. This work provides valuable guidance on the safe and proper application of thiamethoxam for spinach grown in field conditions.

## Figures and Tables

**Figure 1 molecules-27-02209-f001:**
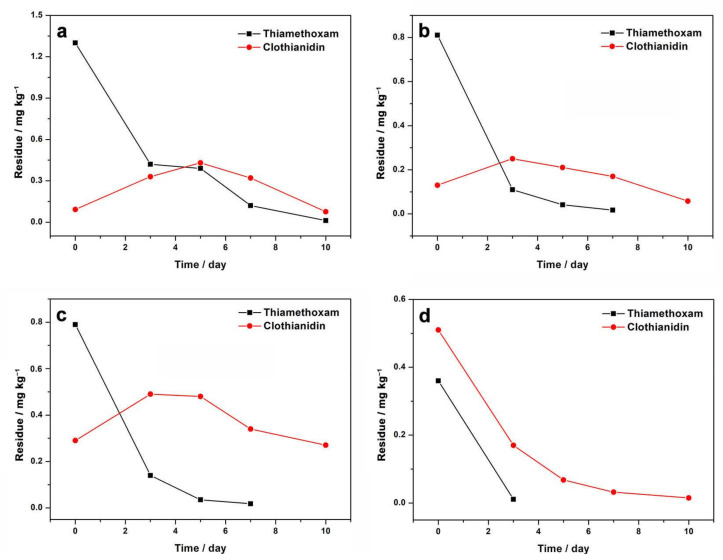
Dissipation of thiamethoxam and clothianidinon spinach samples at Shanxi (**a**), Anhui (**b**), Guangdong (**c**), and Shandong (**d**) sites.

**Figure 2 molecules-27-02209-f002:**
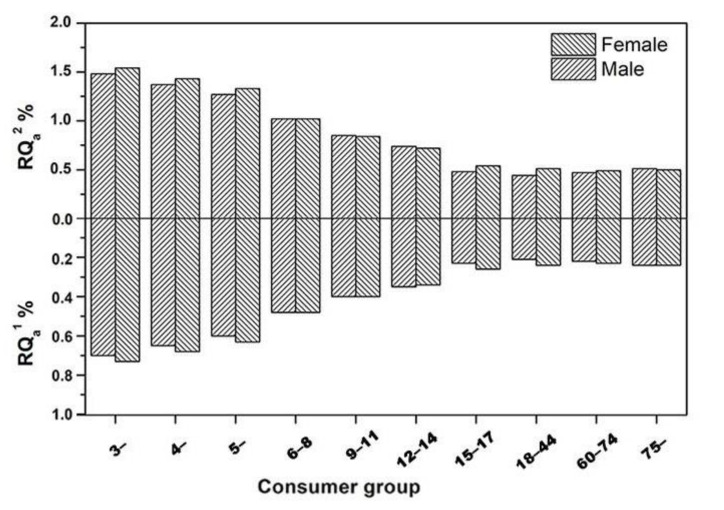
The RQ_a_% of thiamethoxam (RQ_a_^1^) and clothianidin (RQ_a_^2^) residue on spinach for different consumer groups.

**Table 1 molecules-27-02209-t001:** Intra- and inter-day accuracy and precision of thiamethoxam and clothianidin on spinach at four fortified levels.

Analyte	Spiked Level (mg kg^−1^)	Intra-Day (*n* = 5)		Inter-Day (*n* = 15)	
		Accuracy (%)	Precision (% RSDs)	Accuracy (%)	Precision (% RSDs)
Thiamethoxam	0.01	94.5	8.4	102.4	8.6
	0.1	97.3	10.9	96.8	8.1
	2	98.8	8.8	100.7	9.7
	5	98.4	7.5	105.1	9.5
Clothianidin	0.01	96.3	4.6	105.6	8.7
	0.1	101.8	9.3	100.6	7.9
	2	99.1	3.8	103.9	5.9
	5	105.5	6.1	109.7	7.4

**Table 2 molecules-27-02209-t002:** Dissipation dynamic parameters of thiamethoxam on spinach samples at Shanxi, Anhui, and Guangdong sites.

Location	Analyte	Equation	R^2^	Half-Life (Days)
Shanxi	Thiamethoxam	C = 1.854e^−0.44t^	0.914	1.6
Anhui	Thiamethoxam	C = 0.705e^−0.55t^	0.989	1.3
Guangdong	Thiamethoxam	C = 0.736e^−0.55t^	0.987	1.3

**Table 3 molecules-27-02209-t003:** Mean temperature, cultivation facilities, and residue behavior of thiamethoxam during 3 days after application.

Location	Mean Temperature (°C)	Cultivation Facilities	Residue/ mg kg^−1^		Degradation Rate (%)
			0 Day	3 Day	
Shanxi	19.1	Greenhouse	1.3	0.42	67.7
Guangdong	24.5	Shade mesh	0.79	0.14	82.3
Anhui	18.2	Open field	0.81	0.11	86.4
Shandong	26.5	Open field	0.36	0.011	96.9

**Table 4 molecules-27-02209-t004:** Terminal residues of thiamethoxam and its metabolite clothianidin on spinach.

Location	Dosage (g a.i.ha^−1^)	Spray Times	Interval (Days)	Terminal Residue (mg kg^−1^)	
				Thiamethoxam	Clothianidin
Inner Mongolia	27	1	5	0.17, 0.18	0.17, 0.18
			7	<0.010, 0.013	0.043, 0.073
Shanxi			5	0.36, 0.41	0.42, 0.44
			7	0.10, 0.13	0.32, 0.32
Beijing			5	0.041, 0.065	0.17, 0.20
			7	0.018, 0.028	0.12, 0.13
Shandong			5	<0.010, <0.010	0.067, 0.068
			7	<0.010, <0.010	0.030, 0.034
Anhui			5	0.040, 0.041	0.20, 0.21
			7	0.013, 0.021	0.14, 0.19
Hunan			5	0.18, 0.28	0.021, 0.047
			7	0.11, 0.20	0.011, 0.030
Guizhou			5	0.18, 0.22	0.28, 0.28
			7	0.11, 0.16	0.17, 0.38
Guangdong			5	0.029, 0.041	0.43, 0.52
			7	0.017, 0.018	0.33, 0.34

**Table 5 molecules-27-02209-t005:** Acute dietary risk assessment of thiamethoxam and clothianidin on spinach for different consumer groups.

Consumer Group	Bw Kg	LP [37] [g (kg bw·Day)^−1^]	IESTI ^1^ [μg (kg bw d)^−1^]	RQ_a_ ^1^ (%)	IESTI ^2^ [μg (kg bw d)^−1^]	RQ_a_ ^2^ (%)
3– Male	15.8	5.7143	7.0137	0.70	8.8955	1.48
3– Female	15.3	6.0606	7.3084	0.73	9.2692	1.54
4– Male	17.9	5.7143	6.4658	0.65	8.2005	1.37
4– Female	17.2	6.0606	6.7755	0.68	8.5934	1.43
5– Male	20.2	5.7143	5.9963	0.60	7.6051	1.27
5– Female	19.4	6.0606	6.2890	0.63	7.9763	1.33
6–8 Male	25.8	4.8077	4.8316	0.48	6.1279	1.02
6–8 Female	24.6	4.399	4.8036	0.48	6.0924	1.02
9–11 Male	35.8	4.8077	4.0326	0.40	5.1145	0.85
9–11 Female	34.1	4.399	3.9678	0.40	5.0323	0.84
12–14 Male	48.4	4.8077	3.4960	0.35	4.4339	0.74
12–14 Female	45.9	4.399	3.4114	0.34	4.3267	0.72
15–17 Male	57.6	2.3885	2.2605	0.23	2.8670	0.48
15–17 Female	51.5	2.7372	2.5553	0.26	3.2408	0.54
18–44 Male	67.0	2.3885	2.0808	0.21	2.6390	0.44
18–44 Female	56.7	2.7372	2.4238	0.24	3.0741	0.51
60–74 Male	63.9	2.6413	2.2379	0.22	2.8383	0.47
60–74 Female	57.3	2.5042	2.3147	0.23	2.9357	0.49
75– Male	60.5	2.8853	2.4028	0.24	3.0475	0.51
75– Female	53.0	2.4096	2.3804	0.24	3.0190	0.50

Notes: LP: highest large portion provided (97.5th percentile of eaters), data from WHO Food Safety Collaborative Platform (Ref. [37]); RQ_a_: acute dietary exposure risk probability; ^1^ represents thiamethoxam; ^2^ represents clothianidin.

## Data Availability

The data presented in this study are available from the authors on request.

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
