# Peer review of "Dissipation Behavior and Acute Dietary Risk Assessment of Thiamethoxam and Its Metabolite Clothianidin on Spinach"

_molecules, 2022, doi:10.3390/molecules27072209_

Round 1

Reviewer 1 Report

line comment
30 sucking
34 and a major
36 posing a risk to the consumers
41 the residue definition is to consider thiam. and clothiad separately
58 which are these? Can you provide a literature overview of these ?
63 would help to provide guidance
68 provide reference
71 matrix effects. Did you evaluate the matrix effects? Please put the table in the supplementary materials
74 in the supplementary materials
76 express the curves in µg/kg
88-89 Table 1: explain how you quantified the residues at 5 mg/kg using the linear portion of the calibration
106 can you demonstrate that?
107 did you implement a two compartment model? Can you show it?
113 and reached a level lower than the limit of quantitation (LOQ), please always define the terms or provide a reference!
116-117 can you please provide an possible explanation on the figure 1d. Why is that? The study included only application of Thiametoxam, right? No clothiadin was applied in the field, so why the residue at time 2 hr? Did it degrade so fast? 
  did you check the formulation? Did you run a test with the sprayers? I miss an important paragraph about the application process. Were the sprayers clean and calibrated? 
  Figure 1: the individual points are the average results of the two laboratory samples? Can you please provide the data in the supplementary materials?
130 You cannot refer the reader to a table in the supplementary materials and then discuss the content in the paper. Please insert the table in the main text. 
132 resulted in a faster
134 define the cultivation facilities: that is in a greenhouse , in open field or in shaded area
137 can reduce light? It's not the opposite? the more light, the faster the degradation?
138 you need to add a clear sentence to describe the degradation effects and your hypothesis
141 provide a short explanation of what is a terminal study and its objective (provide a reference)

144 what is the MRL in China for thiametoxan and clothiadinin? I looked up a found an MRL for pears and dried chili, but for spinach?
154 Explain the terms IEST and ARfD,U, LP, etc.. this is the first time they appear in the text,paragraph 3.6 is much later in the paper..
156 Explain that you took the Arfd value from literature, this is important!
  In general what HR data did you use?? Can you explain the IESTI using your data? 
160 Explain RQa%, explain what is it or refer the reader to a reference!!!
161 Explain what RQa%=  100 criteria is!! Or provide a reference
162 space!!!n on
162 has no significant effect
164 reword the sentence
  In general I think you need to expand a bit more on paragraph 2.4 and explain better all your data. 
175 explain EW and then add abbreviation
177 what is MerckKGaA>??
182 delete the
187 add space before 0.2
191 according to the
193 included a diss.. and a terminal
199  of the suppl mat
201 ..area to avoid cross contamin
204 what is the total weight of the composite samples
205 were collected
214 are you sure 5 g ??? It can't be
219 after filtration
227 please always refer the reader to the supplementary materials
241 it is a good procedure to express in terms of the LOG and fit the linearity. Did you check that your data fitted the first order ? Can you demonstrate it?
249 you need to explain what are case 2a and 2 b, from where do you take this?. There must be a proper reference and an explanation
261-264 you need to include the bw and the Arfd in the explanations as well
361 improve the citation

Author Response

Thank you for your useful comments and suggestions on the language and structure of our manuscript. We have modified the manuscript accordingly and the detailed corrections are listed in the attachment

Reviewer 2 Report

The part, 3. Material and methods should be before 2. Results and discussion.

Line 144: Please include reference for the MRLs.

Line 266-267: “the residue behavior of thiamethoxam and its metabolite clothianidin 266 on spinach after field application were investigated by the QuEChERS method coupled 267 with LC-MS/MS”. The sentence is wrong, with  QuEChERS - LCMSMS, the residues are determined not the residue behavior.

Line 270: “temperature and light might to be two” please rephrase.

Line 277: “on the reasonable application of thiamethoxam” please rephrase an replace reasonable.

Author Response

We thank the review’s valuable comments. The following are point-by-point response to the review’s comments.

(1) The part, 3. Material and methods should be before 2. Results and discussion.

Response (1): Thanks for your comment. We are sorry that the manuscript was prepared according to the template, in which The part Material and methods is after The part Results and discussion.

(2) Line 144: Please include reference for the MRLs.

Response (2): Thanks for your valuable comment. The “National food safety standard-Maximum residue limits for pesticides in food (GB 2763-2021)” is provided as a reference. The revised part is shown as below:

Line 175: The residue levels of thiamethoxam and clothianidin were in the range of <0.010-0.39 mg kg-1 and 0.021-0.48 mg kg-1 respectively, which were lower than the maximum residue levels (MRLs) prescribed by China[31].

[31] Ministry of Agriculture and Rural Affairs of the People’s Republic of China. GB 2763-2021 National Food Safety Standard-Maximum Residue Limits for Pesticides in Food. Available online: https://www.sdtdata.com/fx/fmoa/tsLibCard/183688.html (accessed on 8 March 2022).

(3) Line 266-267: “the residue behavior of thiamethoxam and its metabolite clothianidin 266 on spinach after field application were investigated by the QuEChERS method coupled 267 with LC-MS/MS”. The sentence is wrong, with QuEChERS - LCMSMS, the residues are determined not the residue behavior.

Response (3): Thanks for your valuable comment. We have revised the sentence, which is shown as below:

Line 323-324: In this study, the residue of thiamethoxam and its metabolite clothianidin on spinach after field application were determined by the QuEChERS method coupled with LC-MS/MS.

(4) Line 270: “temperature and light might to be two” please rephrase.

Response (4): Thanks for your valuable comment. The sentence has been rephrased, which is shown as follows:

Line 327: By comparing dispassion behavior and experiment conditions of four test sites, temperature and light might be two main factors for degradation of thiamethoxam.

(5) Line 277: “on the reasonable application of thiamethoxam” please rephrase an replace reasonable.

Response (5): Thanks for your valuable comment. The reasonable has been replaced and the revised part is shown as follows:

Line 334:This work provides a valuable guidance on the safe and proper application of thiamethoxam for spinach grown in field conditions.

Round 2

Reviewer 1 Report

Thank you for the extensive review. The quality of the paper has really improved.